# Entrepreneurship on Social Networking Sites: The Roles of Attitude and Perceived Usefulness

**DOI:** 10.3390/bs13040323

**Published:** 2023-04-10

**Authors:** Giovanni Di Stefano, Stefano Ruggieri, Rubinia Celeste Bonfanti, Palmira Faraci

**Affiliations:** 1Department of Psychology, Educational Science and Human Movement, University of Palermo, 90128 Palermo, Italy; 2Faculty of Human and Social Sciences, Università degli Studi di Enna “Kore”, 94100 Enna, Italy

**Keywords:** entrepreneurship, social networking sites, perceived usefulness, attitude, technology acceptance model, theory of planned behaviour

## Abstract

Background: Social media platforms are a significant growth opportunity for enterprises, especially for microenterprises, due to the possibility of establishing direct contact with their customers. We investigate the psychological reasons that drive entrepreneurs towards the use of social networking sites (SNSs) for their business, following two important social psychology theories: the theory of planned behaviour and the technology acceptance model. We also tested for two personality traits: openness to experience and dominance. Methods: Data were acquired by examining 325 microentrepreneurs who decided to use either SNSs or traditional sales methods for their businesses. Results and conclusions: Our results confirm that of all the behavioural antecedents tested, perceived usefulness and attitude towards SNSs’ effect on business proved to be the best predictors of the intention to use (or continue using) SNSs for business activity. Implications and suggestions for future research are also discussed.

## 1. Introduction

The world is constantly changing due to the pervasive use of information and communication technologies (ICT) shaping our lives and society. Enterprises have also made extensive employment of ICT in recent years, becoming an indispensable part of entrepreneurship methods, and offering opportunities to reach targets and generate new ideas for businesses. The use of ICT is usually connected with marketing and management strategies, the types of products or services offered, a company’s technological equipment, and so on [1,2,3].

Because of the growing use of social networking sites (SNSs), many studies have expanded previous research on ICT, highlighting how the employment of SNSs platforms allows the increasing of economic performance and survival rates of firms, better internal organisation, and improved communication with customers [4,5,6,7].

Microentrepreneurs (namely, enterprises that employ fewer than 10 persons and whose annual balance sheet total does not exceed EUR 2 million) benefit the most from SNSs, which help them find new contacts [8], establish a business relationship with their customers [9,10], improve decision-making processes, and increase communication with their clients [1,2,11]. The possibility of interacting with customers leads to the creation of a direct and personal relationship that is the basis of customer loyalty [12].

Despite these advantages, many microentrepreneurs are reluctant to use SNSs [13], preferring to use in-store shops and other traditional sales channels to improve their businesses, probably because lower levels of alertness tend to not recognise new entrepreneurial opportunities [14]. Today, little is known about the psychological reasons that could play a role in influencing this choice.

The main aim of our study is to clarify the role of psychological variables in influencing microentrepreneurs’ decision to use SNSs in their business activity. We developed this study using the theory of planned behaviour (TPB) [15] and the technology acceptance model (TAM) [16] as the theoretical framework. We are interested in better understanding the psychological variables that best predict microentrepreneurs’ intention to use SNSs for their own businesses.

## 2. Literature Review and Hypothesis of the Present Study

### 2.1. Entrepreneurship Behaviour and Social Networking Site Usage

Among the many social media, SNSs are considered the most important for entrepreneurs—especially for microentrepreneurs—due to the possibility they present to create interaction, share information with users and groups, and discuss and recruit customers, from whom microentrepreneurs can receive feedback through comments, likes, or reposts [17,18,19]. Through interactions and communications with investors, employees, customers, and suppliers, SNSs offer entrepreneurs a method of low-cost business growth and brand development [20,21]. Despite these economic benefits, many microentrepreneurs believe that using in-store shops (i.e., shops located in the city centre or inside other shopping centres) remains a better model for their businesses [11,22].

In most cases, the choice to use in-store shops rather than make use of new technologies seems dictated by personal preference instead of a careful business plan. In fact, several studies show that when an entrepreneur begins a business activity, they bring their own human capital to the trade, making it an extension of their way of being [23,24]. For this reason, since starting up a new enterprise, the entrepreneur’s personality traits, way of being, and interaction style are essential characteristics in their way of doing business [25,26]. All of these traits point to the importance of a psychological perspective in the study of the use of SNSs by entrepreneurs.

Among the theories that have analysed entrepreneurs’ behaviour, the TPB has provided the best results over the last few years [27,28,29]. Originating from social psychology, the TPB works on the assumption that behavioural intention is the best predictor of behaviour [15]. In the theoretical framework, behavioural intention is the result of three antecedents: attitude (a favourable or unfavourable evaluation of the behaviour), subjective norms (the perceived social pressure to perform or not perform the behaviour), and perceived behavioural control (the perceived ease or difficulty to perform the behaviour) [15].

Many studies have applied this theory to entrepreneurial behaviour. A lot of research has analysed the TPB in relation to the intention to start a business [28,30,31], the intention to innovate [32], and the intention to employ oneself [29]. In previous applications of the theory in the entrepreneurship context, the three independent variables usually accounted for 30–45% of the variance in behaviour intentions [27,28,30,32].

An extension of the TPB in the field of social media is the TAM [16,33,34]. The TAM was created to model users’ acceptance of information technology and to determine the causal relationships among users’ internal beliefs, attitudes, and intentions related to their adoption of technology. It claims that the acceptance of a new technology relates to two beliefs: perceived usefulness and perceived ease of use. *Perceived usefulness* refers to a person’s subjectively assessed probability that using an application system will be helpful in improving performance. *Perceived ease of use* concerns the degree to which a person who uses a particular technological system believes its use would be free of physical and mental effort [16]. Perceived usefulness and perceived ease of use affect an individual’s attitude towards using a technological system [16]. Previous research investigated many aspects of the person’s acceptance of different technologies, such as word processors, the World Wide Web, e-mail, e-learning environments, and e-commerce [35,36,37,38,39].

The integrated model of the TPB and TAM has previously been adopted to study the behavioural intention to use different technologies, such as e-commerce, e-learning, wikis for group work, and others [40,41,42,43,44,45]. Although the predictive power of the TAM was found to be slightly higher than TPB’s [46,47], the TPB was found to provide more helpful information for development than the TAM [48,49]. However, the blended use of the two theories in explaining the uptake of technologies is controversial. For example, the relationship between attitudes and intentions was found to be nonsignificant after controlling for the effect of perceived ease of use and perceived usefulness on intentions [45,50]. Yayla and Hu [51] found that individual theories and analyses of single variables offered more compelling results than integrated theories. This study advances an integrated model of TAM and TPB in an entrepreneurship context, where the relationship between the six variables will supply a reference for a new conceptual model that is integrated by developing eight hypotheses whose results provide an empirical model that describes the behavioural intention to use SNSs for business by attitude toward using.

An entrepreneur’s personality also seems to play a key role in entrepreneurship behaviour and SNS usage, although the results are controversial. Openness to experience has been considered the most influential personality trait in the propensity for innovation [52,53]. People who are highly open to experience are curious, possess imagination and creativity, and are unchained by tradition [53]. Amichai-Hamburger and Vinitzky [54] indicated that people who are highly open to experience were inclined to use several SNS features, likely due to their curiosity and willingness to seek out new experiences. Extraversion is a fundamental trait for entrepreneurs to build social connections, as well. Extraverted entrepreneurs build solid networks to ease their access to external resources, showing dominant traits in social occasions. Additionally, they generally show dominant traits in making decisions that also affect innovation in the entrepreneurial process [55]. Zhao and Seibert [56] emphasised that the sociability of extroverts allows them to mobilise others and develop extensive social interactions, a trait that influences the choice to use different channels of social interaction, such as SNSs [57]. Other research has found that it is not possible to detect personality traits that differentiate entrepreneurs from the rest of the population [58,59,60].

### 2.2. Hypotheses

The main aim of our study was to shed light on the primary differences between entrepreneurs who use SNSs for their business activity and those who do not, focusing on the psychological processes underlying the entrepreneurs’ choices. We evaluated the predictive power of underlying variables of the TPB and TAM in the context of microentrepreneurs’ intentions to use SNSs for their own entrepreneurial activity. Knowing which psychological variables are better predictors of entrepreneurs’ intentions to use SNSs can help understand the entrepreneurial behaviour associated with the use of SNSs and allow the use of broader interpretative models. Figure 1 elaborates on the mixed model on which our hypotheses were grounded.

As a general hypothesis, we expected there to be differences between entrepreneurs who use SNSs for their business activities and those who do not. To test the psychological processes underlying the entrepreneurs’ choices, based on the TPB and TAM, we proposed the following hypotheses:

**Hypothesis** **1.**
*Attitude towards entrepreneurial activity on SNSs positively affects behavioural intention to use SNSs for business.*


According to the TPB, in addition to being determined by one’s attitude towards entrepreneurial activity on SNSs, behavioural intention to use SNSs for business is also influenced by subjective norms and perceived behavioural control. Thus, we expected the following:

**Hypothesis** **2.**
*Subjective norms positively affect behavioural intention to use SNSs for business.*


**Hypothesis** **3.**
*Perceived behavioural control positively affects behavioural intention to use SNSs for business.*


Following the TAM, behavioural intention to use SNSs for one’s business is a function of attitude towards entrepreneurial activity on SNSs, which is, in turn, determined by perceived usefulness and perceived ease of use. Finally, perceived usefulness is a direct determinant of behavioural intention to use SNSs for business, and it is influenced by perceived ease of use. Therefore, we expected the following:

**Hypothesis** **4.**
*Perceived usefulness of SNSs for entrepreneurial activity significantly impacts one’s attitude towards entrepreneurial activity on SNSs.*


**Hypothesis** **5.**
*Perceived ease of use of SNSs significantly impacts one’s attitude towards entrepreneurial activity on SNSs.*


**Hypothesis** **6.**
*Attitude towards entrepreneurial activity on SNSs plays a role in mediating the effect of the perceived usefulness of SNSs on behavioural intention to use SNSs for business.*


Alongside the psychosocial variables so far analysed, and following current research on the relationship between personality factors and entrepreneurs’ attitudes, we further hypothesised:

**Hypothesis** **7.**
*There is a relationship between a high level of openness to experience and one’s attitude towards entrepreneurial activity on SNSs.*


**Hypothesis** **8.**
*There is a relationship between a high level of dominance and one’s attitude towards entrepreneurial activity on SNSs.*


## 3. Materials and Methods

### 3.1. Participants and Procedure

Three hundred twenty-five microentrepreneurs (men = 138, women = 187) between 22 and 68 years old (*M* = 40.34, *SD* = 10.65) participated in this study. The sample was randomly selected from the lists of the local chambers of commerce in three mid-sized cities in southern Italy. All entrepreneurs worked in the field of handmade objects. They produced and traded small jewellery, clothes, home and personal accessories, souvenirs, and gifts. The sample was composed of two subgroups. First were microentrepreneurs who exclusively conducted their business online, especially on Instagram and Facebook, and sold on dedicated marketplaces (e.g., Etsy). We called this group “users” (*n* = 142, men = 56, women = 86). The second group were microentrepreneurs who used in-store shops (i.e., shops located in the city centre or inside other shopping centres). We called this group “nonusers” (*n* = 183, men = 82, women = 101). Both groups were based in the same territory, but the users sold their products online, sending them all over the world; nonusers sold only in their own shops. Microentrepreneurs who used both in-store shops and SNSs as a channel for presenting and selling their products were not enrolled in this study.

After being contacted, participants were asked to respond to an online anonymous questionnaire. The response rate was 68.4% for users and 91.5% for nonusers. This difference was probably because nonusers were approached face-to-face, and users were approached via electronic contact (e-mail or SNSs).

### 3.2. Measures

**Sociodemographic Data.** This study collected the following sociodemographic information: age, gender, number of employees, and revenue of the entrepreneurial activity.

**Attitude towards Entrepreneurial Activity on SNSs.** We assessed participants’ attitudes towards entrepreneurial activity on SNSs with the semantic differential measurement technique [61]. Using a 7-item scale (e.g., bad/good, ugly/beautiful, weak/strong), respondents evaluated the target sentence, “For you, a business activity carried out exclusively through social networking sites is…”. Each item was presented on a 5-point Likert scale (α = 0.90) designed such that the left side was negative term, and the right was positive term.

**Subjective Norms.** Subjective norms were assessed with two items on a 5-point Likert scale (1 = *strongly disagree* and 5 = *strongly agree*; α = 0.75). A sample item for operational definition of variable [62] was “The most significant person for me at this moment (e.g., my partner, one of my parents) believes that the use of social networking sites is a fundamental element to develop my business activity”.

**Perceived Behavioural Control.** We measured behavioural control with five items on a 5-point Likert scale (1 = *strongly disagree* and 5 = *strongly agree*; α = 0.85) based on Ajzen and Fishbein’s operational definition of these variables [62]. We measured users’ perceived behavioural control in working activities on SNSs (e.g., “I think I have all the technological skills for management of my online business led exclusively through SNSs”). We measured nonusers’ perceived behavioural control related to a potential business activity to be carried out on the SNSs (e.g., “I think I have all the technological skills for management of a potential online activity developed exclusively through SNSs”).

**Perceived Usefulness of SNSs in Entrepreneurial Activity.** Using three items adapted from Davis [16], participants were tested on a 5-point Likert scale (1 = *strongly disagree* and 5 = *strongly agree*; α = 0.88). Items included, “I think that social networking sites are useful tools for the purchase of products by modern consumers.”

**Perceived Ease of Use of SNSs.** Perceived ease of use of SNSs was measured using six items adapted from Davis [16] (e.g., “It is easy for me to use social networking sites to find important information”). Each item was presented on a 5-point Likert scale (1 = *strongly disagree* and 5 = *strongly agree;* α = 0.87).

**Dominance and Openness to Experience.** We measured participants’ dominance and openness to experience using the Big Five questionnaire [63,64]. Participants answered 24 questions for each scale (1 = *strongly disagree* and 5 = *strongly agree*; α_dominance_ = 0.84; α_openness to experience_ = 0.81).

**Behavioural Intention to Use SNSs for Business.** Participants’ behavioural intention to use SNSs for business was measured with a single item in a double formulation: for users, the statement was “It is my intention to continue to use social networking sites for my business activity”; for nonusers, the statement was “It is my intention to start using social networking sites for my business activity”. The item was presented on a 5-point Likert scale (1 = *strongly disagree* and 5 = *strongly agree*) based on Ajzen and Fishbein’s operational definition of these variables [62].

## 4. Results

Data were analysed using SPSS, version 22. We conducted a preliminary examination of the variables’ distribution to assess the extent to which the data could be analysed using normal theory estimation procedures. Inspection of the standardised variables’ distributions revealed some univariate outliers. Accordingly, 12 cases were removed from the dataset. Nevertheless, Mardia’s multivariate kurtosis index (Mardia = 85.36) still indicated some departures from normality. Hence, based on the Mahalanobis distance estimation, three multivariate outlier cases were excluded from successive analyses. Further, applying Cook’s distance analysis, one influential data point was identified and dropped out. Overall, regression diagnostics revealed the presence of 15 cases to be excluded from the analyses, which yielded a final sample of 310 participants.

A general linear model analysis was applied to verify the differences in the main variables of this study between users and nonusers (see Table 1).

The results were in line with expectations, showing that the two groups were similar in terms of sociodemographic characteristics, but differed in all other variables except for dominance. In particular, the SNS users displayed a more positive attitude towards entrepreneurial activity on SNSs than nonusers. SNS users also scored higher in levels of subjective norms, perceived behavioural control, perceived usefulness of SNSs in their entrepreneurial activity, and perceived ease of use of SNSs. Finally, users had greater openness to experiences, but did not differ in levels of dominance.

### 4.1. Hierarchical Regression Analyses on Behavioural Intention

A four-step hierarchical regression analysis was conducted to test H1, H2, and H3. In the first step, gender and age were introduced to control possible confounding effects. In the second, third, and fourth steps, attitude, subjective norms, and perceived behavioural control, respectively, were added to detect the main effects on behavioural intention. The significance of the change in squared multiple correlations was assessed at each step.

When regression analysis was conducted with behavioural intention as a dependent variable, gender accounted for a significant proportion of the explained variance (∆*R*^2^ = 0.06, ∆*F* = 9.77, *p* < 0.001), contributing significantly to the prediction of behavioural intention (*β* = 0.11, *p* < 0.05). Women showed higher attitude scores than men (*F* = 4.53, *p* < 0.05). Age negatively predicted behavioural intention scores (*β* = −0.21; *p* < 0.001). In addition, our results showed that attitude positively predicted behavioural intention (*β* = 0.53, *p* < 0.001). Further, subjective norms and perceived behavioural control positively predicted behavioural intention as well (*β* = 0.24, *p* < 0.001; *β* = 0.12, *p* < 0.05, respectively). The results of the regression models are reported in Table 2.

These data confirmed H1, H2, and H3, showing how the TPB is a useful framework for understanding the antecedent of behavioural intention to use SNSs for business.

To better understand these relationships, two additional four-step hierarchical regression analyses were conducted to test H1, H2, and H3 among users and nonusers. As shown in Table 2, attitude and perceived behavioural control were the two significant predictors of the behavioural intention to use SNSs for business in both groups. This was the same as in the overall condition, although perceived behavioural control had a lower level of significance.

A further moderated hierarchical regression analysis was conducted to evaluate the contribution of TAM and personality variables. In the opening hierarchical step, gender and age were introduced to control possible confounding effects. In the second and third steps, the perceived usefulness of SNSs to entrepreneurial activity (H4) and perceived ease of use of SNSs (H5) were added to detect the main effects of attitude towards entrepreneurial activity on SNSs. In steps four and five, the predictor variables openness to experience (H6) and dominance (H7) were inserted to examine their association with attitude (see Table 3).

Gender, which was introduced as a control variable in the first step of the hierarchical analysis, accounted for a significant proportion of the explained variance (∆*R*^2^ = 0.04, *F* = 5.65, *p <* 0.01). Gender itself also contributed significantly to the prediction of attitude (*β* = 0.12; *p* < 0.05). Females showed higher attitude scores than males (*F* = 4.36, *p* < 0.05). Likewise, age negatively predicted attitude levels (*β* = −0.14, *p* < 0.05). Additionally, perceived usefulness and perceived ease of use were both positively predictive of participants’ attitudes towards entrepreneurial activity on SNS scores (*β* = 0.75, *p* < 0.001; *β* = 0.15, *p* < 0.001, respectively), whereas dominance and openness to experience were not significantly predictive of attitude (*β* = 0.01, ns, and *β* = −0.02, *ns,* respectively). The main effects model for the whole group is depicted in Figure 2.

Once again, users and nonusers were separately tested. We observed the same relational pattern (see Table 3), with main effects of both perceived ease of use and perceived usefulness (with a stronger effect of perceived usefulness).

We also tested for the effects of dominance and openness to experience. Among users, neither variable affected the attitude towards entrepreneurial activity on SNSs. Among nonusers, only openness to experience exhibited a weak effect.

### 4.2. Mediation Analysis

According to Baron and Kenny’s [65] causal steps method, we conducted a path analysis to verify whether attitude levels mediated the relationships between perceived usefulness and the behavioural intention to use SNSs for business. 

Based on our findings, perceived usefulness was significantly associated with behavioural intention (*β* = 0.55, *t* = 11.44, *p* < 0.001) and attitude (*β* = 0.76, *t* = 20.61, *p* < 0.001). After controlling for attitude, the relationship between perceived usefulness and behavioural intention was still significant (*β* = 0.30, *t* = 4.25, *p* < 0.001), but lesser than the overall effect of perceived usefulness on behavioural intention. Thus, attitude only partially mediated the perceived usefulness–behavioural intention relationship (see Figure 3).

## 5. Discussion

The main aim of our study was to better understand the psychological antecedents that best predict microentrepreneurs’ intention to use SNSs for their own businesses, examining the goodness of fit of the hypothesised structural model by integrating TPB and TAM.

Among the first variables observed in this research were age and gender. Women showed more positive attitudes towards entrepreneurial activities on SNSs than men. This is in line with recent studies that showed how women are progressively improving the way they use social media for their work activities, to the point of overtaking men [66,67]. Likewise, as was easy to expect, greater age negatively predicted attitudes towards the use of SNSs for business. Younger people showed a more favourable attitude towards the use of SNSs for business than most aged people did, and this trend is supported by a lot of evidence in the literature [68,69].

Differences between users and nonusers also emerged. Although our participants did not differ significantly in age, gender, number of employees, revenue, or product category, users intended to continue using SNSs more than nonusers. Users also exhibited a more favourable attitude towards SNSs in their businesses, greater perceived behavioural control, greater perception of utility, and so on. All this was consistent with participants’ behaviour to use or not use SNSs in their business activity, in line with the TPB. Higher values of these variables were associated with the behaviour of SNS use for business (users) and, on the contrary, lower values with its rejection (nonusers).

Analysing the antecedents of participants’ choices of whether to use SNSs for their own businesses, it was possible to detect how all the variables investigated in this study were good or excellent predictors of entrepreneurs’ intention to use SNSs for business, supporting the use of the TPB and TAM theories in the field of entrepreneurship research.

Referring to the TPB indicators, our results globally showed a significant association with all the variables of the model. Attitude, subjective norms, and perceived behavioural control influenced microentrepreneurs’ intention to use SNSs for business as expected. Attitude was the most influential variable on behavioural intention to use SNSs among the three TPB variables. As observed in previous research, attitude is not always the best predictor of behavioural intention in the TPB. For example, in cases of Internet purchasing behaviour or switching intentions towards public transit, social norms were the most important variable [42,45]. In a study of students’ intentions to use a wiki for group work and their behaviours in doing so, perceived behavioural control was the best predictor [43]. In our study, attitude towards SNSs for business use was the variable that predicted behavioural intention to use SNSs for one’s own business better than others. Other research has observed how the decision to adopt SNSs in an entrepreneur’s business is influenced by the observation of customers and competitors, but above all, from one’s attitude to these media and beliefs about how useful they can be for one’s business [70]. The reason is probably connected with the specificities of entrepreneurs, who are highly autonomous and determined in their decisions, hardly resorting to external suggestions for their choices and thinking to control everything that happens around them [31,71]. In other studies, it has also been observed that entrepreneurs’ attitudes are also extremely important in areas such as risk taking [72] and decision making [73].

Related to perceived behavioural control, it is known that entrepreneurs tend to overestimate their ability to control the environment and the objects therein. Ahmad [74] clearly explained that entrepreneurs are capable of minimising the negative impact of the business environment if they always equip themselves with the necessary competencies. Even with respect to SNSs, entrepreneurs believe they can possess the requisites that will allow them to move easily in the world of SNSs.

We also analysed the precursors of attitude towards entrepreneurs’ activity on SNSs: dominance; openness to experience; and, following the TAM, the perceived usefulness of SNSs in their entrepreneurial activity and perceived ease of use of SNSs.

Unlike what we hypothesised, dominance and openness to experience did not prove to be good predictors of entrepreneurs’ attitudes towards business activity on SNSs, contradicting some of the literature on this research topic. Usually, people who score high on openness to experience tend to be adventurous and creative, less traditional, intellectually curious, and have an overall greater tolerance for unfamiliar things [75]. Additionally, past research has shown that openness to experience is a significant predictor of social media usage [52,53]. Although we confirmed that users had higher levels of openness to experience than nonusers, it was not possible to detect dominance or openness to experience as good predictors of the attitude towards entrepreneurial activity on SNSs.

The perceived usefulness of SNSs in entrepreneurial activity and perceived ease of use of SNSs were also tested, revealing that both had a role in determining entrepreneurs’ attitudes towards the use of SNSs for business. Again, previous research has found that both can play a prominent role under various circumstances. Thus, in some cases, the best antecedent of the attitude was perceived usefulness, as in the case of the intention to use e-government services [76]. In case studies of the ability to use SNSs to monitor one’s partner [39] or students’ willingness to shop online [77], the best antecedent was perceived ease of use.

We found a much stronger association with the attitude in the case of perceived usefulness among users and nonusers. The reason is probably connected with entrepreneurs’ constant search for opportunities [78,79]. Those who see an opportunity in using SNSs for their business tend to have a more positive attitude towards the same tools. On the contrary, those who consider the impact of SNSs low in their business show a less favourable attitude towards the use of SNSs.

To better understand the relationship between behavioural intention to use SNSs for business, attitude towards entrepreneurial activity on SNSs, and perceived usefulness of SNSs in entrepreneurial activity, we tested a mediation model. The results suggested that perceived usefulness plays a fundamental role in the interpretation of this relationship, exerting a direct effect on the intention of behaviour and an effect mediated by attitude. The role of this variable is, therefore, decisive in the intention to use, or continue to use, SNSs for business. This specificity is probably connected with one of the fundamental characteristics of entrepreneurs: opportunity competency, or the capability to recognise market opportunities through several means [80,81,82,83]. This competency is one of the more important and distinguishing competencies for successful entrepreneurs [84]. It represents the ability to search for, recognise, develop, and evaluate all possible opportunities available in a certain market [85]. Only by recognising opportunities can entrepreneurs achieve success in their businesses and avoid potential risks [82,86,87]. Specifically, those who perceived the usefulness of SNSs changed their previous attitudes towards the use of SNSs in their business activities. When attitudes did not change, the perceived usefulness exerted a direct effect on the intention to use SNSs for one’s business. Those who did not perceive the usefulness of the SNSs did not change their attitudes and maintained the firm intention not to use them in their business.

This study has several strengths. Firstly, it enriches the literary landscape of the integrated model of TPB and TAM in the context of entrepreneurship by highlighting the different burdens that variables belonging to the two models have in predicting the intention to use SNSs for one’s business. Secondly, it brings together two different streams of literature, personality traits, and TPB and TAM models to derive hypotheses that can explain the intention to use SNSs for business activities by entrepreneurs. Finally, through these empirical findings, it is possible to identify a psychological profile of micro-entrepreneurs who have a predisposition to using SNSs for their business, useful for market strategies in the business innovation sector.

## 6. Limitations and Future Research

Our study had certain limitations. First, it relied on the participants’ self-reports of psychosocial variables and might be susceptible to response tendencies such as social desirability. Future research should make every effort to employ behavioural measures to overcome such potential biases. Second, although we used a randomised sample from a reference population, we used a small number of participants from a single territorial area, which may not be an accurate representation of the overall population. It is possible that the comparison between different territorial realities—especially from the point of view of economic development—could provide further elements to investigate the relationships tested in this study. Finally, our study focused only on microentrepreneurs in the field of handmade objects. Further analysis should investigate the model proposed here in small- and medium-business entrepreneurs, but also in different product categories.

## 7. Conclusions

Overall, our study contributed to shedding light on the complex and poorly studied relationship between entrepreneurs and SNSs. What emerged is the extremely important role of the variables of the TPB (attitude, perceived behavioural control, and subjective norms) and of the TAM (perceived ease of use and perceived usefulness) in explaining this relationship. Based on this research, a particular role is assigned to attitude and perceived usefulness. On the contrary, personality traits are of secondary importance in influencing the choice to use SNSs for one’s business.

All of this has very important practical implications. SNS-based e-commerce provides opportunities for growth never seen in the past. Perceiving the usefulness of SNSs as a tool for improving one’s business is, in fact, the first of the useful elements to change personal attitude and, therefore, to increase the desire to use SNSs with conviction in one’s daily work. Entrepreneurs’ propensity to innovation is more suitable for sustaining businesses in an unpredictable and turbulent environment like the current one [88], allowing their employees to be future-oriented, self-confident, and experience less stress and job insecurity [89,90,91] by utilising resources available and sustaining employee resilience in the face of adversity. 

## Figures and Tables

**Figure 1 behavsci-13-00323-f001:**
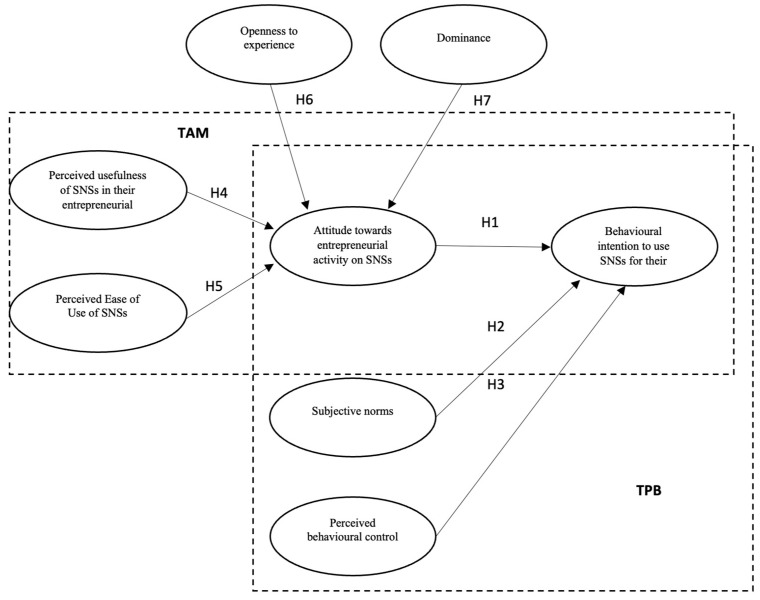
The research model.

**Figure 2 behavsci-13-00323-f002:**
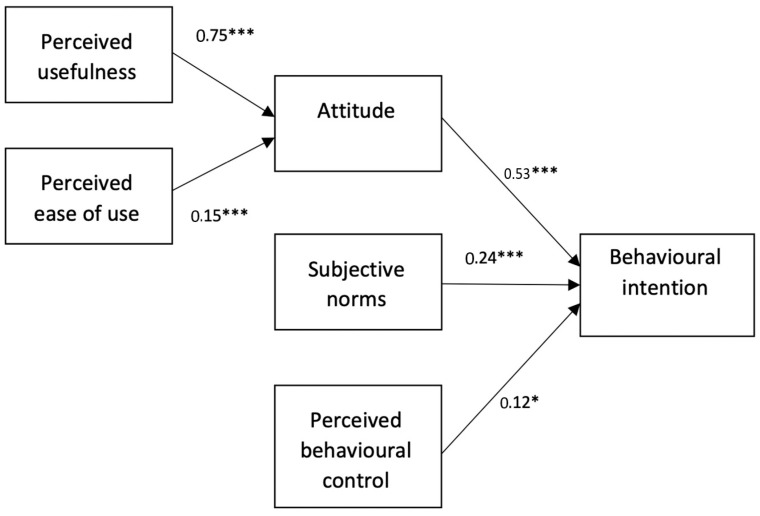
Main effects model. Note. * *p <* 0.05; *** *p <* 0.001.

**Figure 3 behavsci-13-00323-f003:**
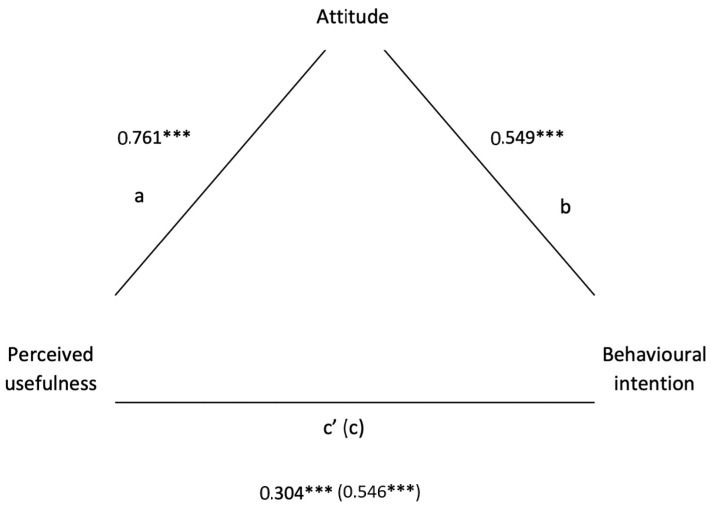
Mediation diagram (perceived usefulness⇒attitude⇒behavioural intention). Note. *** *p* < 0.001; a = effect of Perceived usefulness on Attitude, b = effect of Attitude on Behavioural intention, c = total effect, c’ = direct effect.

**Table 1 behavsci-13-00323-t001:** Differences in the main variables of this study between users and nonusers.

	Users	Nonusers	F	*p*	Partial η^2^
Age	40.78	39.79	0.675	0.415	0.002
Number of employees	1.60	1.63	0.023	0.880	0.001
Revenue	27198	29868	1.669	0.198	0.009
Attitude towards SNSs	3.88	2.98	74.96	0.000	0.196
Subjective norms	4.10	3.72	12.41	0.000	0.039
Perceived behavioural control	3.32	3.02	7.58	0.006	0.024
Perceived usefulness of SNSs	4.20	3.44	40.93	0.000	0.117
Perceived ease of use	4.33	3.73	43.49	0.000	0.124
Dominance	37.47	37.23	0.11	0.738	0.000
Openness to experience	47.52	42.17	64.46	0.000	0.173
Behavioural intention to use SNSs	4.27	3.58	29.65	0.000	0.088

**Table 2 behavsci-13-00323-t002:** Hierarchical regressions of behavioural intention on attitude, subjective norms, and perceived behavioural control.

		*β*	T	*R* ^2^	∆*R*^2^	∆*F*
Overall	Step 1. Gender	0.112	2.018 *	0.060	0.060	9.767 ***
	Age	−0.213	−3.834 ***			
	Step 2. Attitude	0.528	11.024 ***	0.329	0.268	121.522 ***
	Step 3. Subjective norms	0.244	4.635 ***	0.373	0.044	21.487 ***
	Step 4. Perceived behavioural control	0.120	2.172 *	0.383	0.010	4.717 *
Users	Step 1. Gender	−0.034	−0.390	0.026	0.026	1.716
	Age	−0.159	−1.835			
	Step 2. Attitude	0.285	3.421 ***	0.106	0.080	11.701 ***
	Step 3. Subjective norms	0.002	0.024 *	0.106	0.000	0.001
	Step 4. Perceived behavioural control	0.148	1.654	0.125	0.019	2.735
Nonusers	Step 1. Gender	0.173	2.367 *	0.089	0.089	8.361 ***
	Age	−0.240	−3.283 ***			
	Step 2. Attitude	0.535	8.476 ***	0.360	0.271	71.846 ***
	Step 3. Subjective norms	0.357	5.140 ***	0.446	0.087	26.418 ***
	Step 4. Perceived behavioural control	0.140	1.761	0.456	0.010	3.100

* *p <* 0.05; *** *p <* 0.001.

**Table 3 behavsci-13-00323-t003:** Hierarchical regressions of attitude on perceived usefulness and ease of use.

		*β*	*t*	*R* ^2^	∆*R*^2^	∆*F*
Overall	Step 1. Gender	0.117	2.075 *	0.036	0.036	5.653 **
	Age	−0.144	−2.559 *			
	Step 2. Perceived usefulness	0.753	19.922 ***	0.583	0.547	396.901 ***
	Step 3. Perceived ease of use	0.148	3.464 **	0.599	0.016	12.001 ***
	Step 4. Dominance	0.006	0.165	0.599	0.000	0.027
	Step 5. Openness to experience	−0.015	−0.407	0.599	0.000	0.166
Users	Step 1. Gender	0.038	0.432	0.009	0.009	0.594
	Age	−0.084	−0.966			
	Step 2. Perceived usefulness	0.809	15.625 ***	0.656	0.647	244.146 ***
	Step 3. Perceived ease of use	0.110	2.096 *	0.667	0.011	4.395 *
	Step 4. Dominance	0.022	0.403	0.667	0.000	0.162
	Step 5. Openness to experience	−0.091	−1.700	0.675	0.007	2.889
Nonusers	Step 1. Gender	0.148	1.976 *	0.052	0.052	4.669 **
	Age	−0.172	−2.306 *			
	Step 2. Perceived usefulness	0.669	11.680 ***	0.475	0.423	136.432 ***
	Step 3. Perceived ease of use	0.105	1.505	0.482	0.007	2.265
	Step 4. Dominance	0.029	0.477	0.483	0.001	0.227
	Step 5. Openness to experience	−0.121	−2.149 *	0.497	0.014	4.619 *

* *p* < 0.05; ** *p* < 0.01; *** *p* < 0.001.

## Data Availability

Data are available upon request.

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
