# Peer review of "Entrepreneurship on Social Networking Sites: The Roles of Attitude and Perceived Usefulness"

_behavsci, 2023, doi:10.3390/bs13040323_

Round 1

Reviewer 1 Report (Previous Reviewer 3)

2.1. Entrepreneurship Behaviour and Social Networking Site Usage

Referring to the green color mark, this part needs to explain clearly the present study of a theoretically integrated model of TAM and TPB in the entrepreneurship context, I suggested that each variable of the present study should be defined which sounds different from the previous study. Otherwise, didn’t make sense of integrated the previous study if the definition is the same, what you have variables in this study and belong to which, TAM or TPB model in the present study, to make sure this sees your proposed model design or Figure 1. The research models.

2.2. Hypotheses Developments

If you write in this way people will confuse, people didn’t clear relationship between the variable. (To test the psychological processes underlying the entrepreneurs’ choices, based on the TPB and TAM,).

The H4-H6 is very confusing, please refer to your research model, and explain clearly.

H7-H8 – make sure which is higher level and which is lower level, because of confusion.

3.2. Measures

Dominance and Openness to Experience.

Behavioral Intention to Use SNSs for Business

Both of these didn’t clearly need to revise.

Results

The model, hypothesis, and results of the study are completely different.

For what the author designed the model and hypothesis if they didn’t analysis the data according to the hypothesis that make confusion.

Note: You could analyze the gender separately which could see the results of nonusers and or the gender user of in-store shops.

Then analysis the data according to the research model and hypothesis.

With Smart-PLS. that will make it clearer and make sure the mediating effect, doesn’t confuse with moderating effects. As seen in the model you didn’t have the moderating effects but the controlling variable.

2023/2/9

Author Response

Reviewer 2 Report (Previous Reviewer 2)

Introduction

The author has tried to accommodate input regarding the research problem statement previously presented. The researcher writes as follows

"Despite these advantages, many microentrepreneurs are reluctant to use SNSs [13], preferring to use in-store shops and other traditional sales channels to improve their business. This is probably because people with lower levels of alertness tend not to recognize new entrepreneurial opportunities [14]. Today, little is known about the psychological reasons that could play a role in influencing this choice."

However, the revisions submitted, especially with the use of reference number 14, are derived from research 17 years ago. How can information / research results from 17 years ago still be considered relevant today?

What data or what references are used by the researcher so that the researcher can write the following statement:

“Today, little is known about the psychological reasons that could play a role in influencing this choice”.

Research Methods

Write down specifically the sampling technique used. It was written that the respondents were randomly selected. What kind of random is used? (e.g., simple random, systematic random, etc.). Researchers also have not accommodated input from reviewer regarding the need to provide justification for the number of samples used.

Results

Whatever the results of the hypothesis testing carried out are meaningless if the reliability and validity of the research indicators used cannot be accounted for. Reliability and validity are required to ensure that the research indicators used are reliable and measure what should be measured. The reasons put forward by researchers about only assessing reliability and not validity are not easy to understand. Reliability is a requirement for validity. A good study will convey reliability and validity results to show readers that the indicators used are appropriate. The reason the indicators used are adapted or borrowed from previous studies does not guarantee that the indicators are reliable and valid.

Round 2

Reviewer 1 Report (Previous Reviewer 3)

Results in Table 2. Check again some missed in the table 

And Variable... intention also missed. 

Table 3. ( Description in this table is missing and did not match with the model of the study. 

Note: The study results still use the data analysis technique very traditionally. And the problem is that the results of the study are missing the hypothesis and didn't match with the model.

Suggestions: please check again your hypothesis and the model and the results in the table and followed sentence about the results of the study.

Otherwise hard to approve the results of the study. 

2023/2/23

Author Response

It is hard for us to respond to the Reviewer's comments as s/he return, once again, to issues we think we have already discussed. Indeed, the comments contain observations that are, in our opinion, wrong.  

*(1)**Results in Table 2. Check again some missed in the table.*
S/he probably refers to R^2 , ∆R^2 and ∆F that are missing in line of “Age” variable. Being the variables “Age” and “Gender” introduced together at the first step of the regression analysis the values are the same and it doesn't make sense repeat them again in the second line (in every Table that describe a Hierarchical Regression Analyses you can observe this).  

*(2)**And Variable... intention also missed.*  
“Behavioural intention to use SNSs” is our dependent variable. Is s/he saying that we should have regressed the dependent variable with itself?!?!?!?!?! What a nonsense. Data related to “Behavioural intention to use SNSs” are all reported in Table 1.  

*(3)**Table 3. ( Description in this table is missing and did not match with the model of the study.*  
The description is entered correctly, and is identical to the one in the previous table (which was not denied). The form is the classic one which is reported in a hierarchical regression "effect of independent variables on dependent variable” (in our paper “Table 3. Main Effects of Perceived Usefulness and Perceived Ease of Use on Attitude.”). Obviously this makes the exact match with the hypotheses of the study.  

*(4)**Note: The study results still use the data analysis technique very traditionally.*  
Would that be a negative? and if so, where? we don't understand what s/he means.  

*(5)**And the problem is that the results of the study are missing the hypothesis and didn't match with the model.*
S/He keeps reiterating the point without providing guidance. We have formulated some hypotheses (and there has been no dispute about those), we have added others (and there has not been any dispute about this either). We tried to modify the model by reducing the importance of TPB and TAM and focusing more on the variables inspired by them, and it continues to say that "results of the study are missing the hypothesis and didn't match with the model". We really don't understand.

*(6)**Suggestions: please check again your hypothesis and the model and the results in the table and followed sentence about the results of the study.*  
We tested our hypotheses and added more. The hypotheses are all formulated in the classical form "The effect of IV on DV". The effect is revealed using a regression model. The effects of IVs on DV were then analyzed in the Discussion section. But is his/her a dispute on the data analysis technique we applied? But s/he doesn't suggest any other...  How would s/he intend to evaluate the effects of one variable on another other than with a regression?

Round 3

Reviewer 1 Report (Previous Reviewer 3)

Comments and suggestions to Authors

Seeing the Research model, the Hypothesis and results of the study didn’t have reasonable effects on the research model

Hypothesis

Note 1: Hypothesis 8 – See the research model that could be direct and indirect effects on behavioral intention to use SNSs for business. Refer to more suggestions below for revision

Note 2: Hypothesis 9-12 didn’t make sense with the research model and also with the results of the study.

Suggestions 1: The Author could make H10 a moderating effect on Users and Nonusers

Results of study

Suggestion 1: The author should analyze the Sociodemographic Data first and then report the results

Suggestion 2: The Authors should analyze the main research model structures and then report the validation of the model.

Suggestion 3: The Authors should analyze the main research model structures and then report the results of each relationship affecting H1—8 and 9 as a mediation effect.

Suggestion 4: The results of Table 1,2,3, you need to revise then could use them to report the result of the moderating effect of Users and Nonusers.

4.1. Hierarchical Regression Analyses

Table 2.

Note 1: Gender and age were introduced to control possible confounding effects and test H9 and H11 – Alright

Note 2: In the second, third, and fourth steps, attitude, subjective norms, and perceived behavioral control, respectively, were added to detect the main effects on behavioral intention.

Question: where is data analysis of behavioral intention

Table 3. Main Effects of Perceived Usefulness and Perceived Ease of Use on Attitude

Question: What data is your analysis in step 5?

So, make confusion results in Table 2 and Table 3.

I concluded that to revise this manuscript see carefully the note and suggestion and your model.

After reading all the results of your study Table 1, 2, and 3 are just the results of moderation effects of Users and Nonuser and the research model has no meaning.

2023/3/2

Author Response

As we already wrote in our previous reply, we do not agree with the reviewer's comments.
We have already explained the reasons in previous notes.
We defer to academic editor to make a final decision.
Thank you for your time.

This manuscript is a resubmission of an earlier submission. The following is a list of the peer review reports and author responses from that submission.

Round 1

Reviewer 1 Report

This article is excellent with no more suggestion.

Reviewer 2 Report

Introduction

Research problems are not conveyed with the support of data or references, only written "Despite these advantages, many microentrepreneurs are reluctant to use SNSs, preferring to use in-store shops and other traditional sales channels to improve their business. Today, little is known about the psychological reasons that could play a role in influencing this choice".

Literature Review

The development of the hypothesis, especially for the dominance variable, seems to have suddenly appeared in the narrative of developing the hypothesis. There is no explanation and writing of the "dominance" variable in the narrative of developing the hypothesis, but only written in the research hypothesis (H7). There is no information why dominance is used as a variable that represents personality.

In the results section, especially sub-section 5.2 regarding mediation analysis. Why are there results of analysis regarding mediation when there is no research hypothesis indicating the existence of a mediation role.

Methods

·       There is no mention of the sampling technique and justification for the number of samples used.

·       Not all research indicators are explained where they come from: whether they were self-developed or derived from previous research (for example indicators of attitude, subjective norms, perceived behavioural control, and intention).

Results

What are the reasons the researchers did not test the validity of the research indicators used? It is possible if there is a conclusion that the results of testing the hypothesis become meaningless if the research indicators used are doubtful.

Reviewer 3 Report

Comments and suggestions for Author 

Abstract

In the abstract first the numbering in the beginning,

Suggestions: don’t need to put number in.

Introduction

The aim or the purpose of study didn’t clear, what problem the Author want to recover from this study. See the page 2 of 15 RL 40-52, RL and 40-41, 42-44 references

2. Entrepreneurship Behaviour and Social Networking Site Usage

Before this title (Entrepreneurship Behaviour and Social Networking Site Usage), could be add the title of literature review and Hypothesis

The integrated model of the TPB and TAM in this study didn’t explain clearly, why the Author want to use this theory in present study and the definition of present study also didn’t clear.

Page 2 of 15, RL 72-73 and 78 references?

RL 80-83 sentence – statement a little could be misunderstand.

3. Hypotheses

Hypothesis 6 and 7 I couldn’t understand, please check with the model, what you want to explained in this Hypothesis very confusion.

4. Materials and Methods

The didn’t explaining what tool they use for data analysis.

5. Results

Table 1. 6 of 15, RL 246-247 Please check again.

And In RL 244 – 245 - perceived behavioural control?

Table 1. Differences of the main variables of the study between users and nonusers, In RL 239

The results of age analysis is strange and where is the gender?

Similar problem in another results in Table 2,3.

5.1. Hierarchical Regression Analyses on Behavioural Intention? RL 248

The result of study introduced like a messed up which hard to understand by the reader for example:

Table 2. Hierarchical regressions of Behavioural Intention on Attitude, Subjective Norms and Perceived Behavioural Control? It’s confusing which didn’t match to model.

Table 3. D ? Hierarchical regressions of Attitude on Perceived Usefulness and Ease of Use.

Page: 7 of 15 RL 270-271- These data confirmed H1, H2, and H3, showing how the TPB is a useful framework for understanding the antecedent of behavioural intention to use SNSs for business (confusion the model measurements and design)

Page: 7 of 15 RL 275-277 and with table 2 check and rewrite

Page 7 of 15 RL 286- 295 and such also in Figure 2 check and rewrite

Just a sample: whereas dominance and openness to experience were not significantly predictive of attitude (β = .01, ns, and β = −.02, ns, respectively)., there is in Figure 2

Note: The results

1.    The result of the study unreasonable in data collection design the Author mention to measurements the gender men and female and also age but the results of study didn’t show of these results, the validity of data analysis is not very significantly approved.

2.    The study either it moderating nor measuring the structure of the model, the results is very confusing the reader.

3.    The Autor mentioned the mediation results but there isn’t hypothesis on that’s, why the Author want to do of meditating effects? And why only that variable? Suggestions: The author read more regarding meditation relationship, and of the variable effects.

4.    6. Discussion

Many parts in this discussion need to revise, just one example – RL 333-339 check with results of study.

Note: please related the explanation to the results of study and the contribution of present study

8. Conclusions also needs to revise many of the explanation didn’t relate to present study.

Good luck

2022/12/8